# Modified Crosstalk between Phytohormones in Arabidopsis Mutants for PEP-Associated Proteins

**DOI:** 10.3390/ijms25031586

**Published:** 2024-01-27

**Authors:** Ivan A. Bychkov, Aleksandra A. Andreeva, Radomira Vankova, Jozef Lacek, Natalia V. Kudryakova, Victor V. Kusnetsov

**Affiliations:** 1Timiryazev Institute of Plant Physiology, Russian Academy of Sciences, Botanicheskaya 35, Moscow 127276, Russia; ivan.a.b@mail.ru (I.A.B.); alexaa27@mail.ru (A.A.A.); vkusnetsov2001@mail.ru (V.V.K.); 2Institute of Experimental Botany, Academy of Sciences CR, Rozvojova 263, 165 02 Prague, Czech Republic; vankova@ueb.cas.cz (R.V.); lacek@ueb.cas.cz (J.L.)

**Keywords:** *Arabidopsis thaliana*, phytohormones, gene expression, PEP-associated proteins, mutants

## Abstract

Plastid-encoded RNA polymerase (PEP) forms a multisubunit complex in operating chloroplasts, where PEP subunits and a sigma factor are tightly associated with 12 additional nuclear-encoded proteins. Mutants with disrupted genes encoding PEP-associated proteins (PAPs) provide unique tools for deciphering mutual relationships among phytohormones. A block of chloroplast biogenesis in *Arabidopsis pap* mutants specifying highly altered metabolism in white tissues induced dramatic fluctuations in the content of major phytohormones and their metabolic genes, whereas hormone signaling circuits mostly remained functional. Reprogramming of the expression of biosynthetic and metabolic genes contributed to a greatly increased content of salicylic acid (SA) and a concomitant decrease in 1-aminocyclopropane-1-carboxylic acid (ACC) and oxophytodienoic acid (OPDA), precursors of ethylene and jasmonic acid, respectively, in parallel to reduced levels of abscisic acid (ABA). The lack of differences in the free levels of indole-3-acetic acid (IAA) between the *pap* mutants and wild-type plants was accompanied by fluctuations in the contents of IAA precursors and conjugated forms as well as multilayered changes in the expression of IAA metabolic genes. Along with cytokinin (CK) overproduction, all of these compensatory changes aim to balance plant growth and defense systems to ensure viability under highly modulated conditions.

## 1. Introduction

The regulation of organellar gene expression in plants is principally performed by proteins encoded by nuclear genes [1]. Of these, components of the plastid transcription machinery play a fundamental role in fine-tuned regulation of the activity of the plastid genome to adjust it to developmental and environmental changes. This phenomenon is especially relevant for the transcriptional response to phytohormones, which harmonize the expression of nuclear and plastid genomes under various challenges.

In *Arabidopsis*, the specific changes in plastid genome expression are governed by three plastid RNA polymerases. Two monomeric nuclear-encoded phage-type RNA polymerases (NEPs), RPOTp and RPOTmp, are mainly responsible for the transcription of housekeeping genes. The third one, plastid-encoded eubacterial type RNA polymerase (PEP), transcribes photosynthesis genes. PEP activity is regulated by 6 nuclear-encoded sigma transcription factors and requires a set of PEP-associated proteins (PAPs) essential for the proper assembly and/or stability of the PEP complex [2]. Research over the last decade has revealed that PAPs are highly diverse in terms of their activities and structure and contain functional groups involved in DNA/RNA metabolism, redox regulation and ROS protection [3]. However, independent of their function, PAPs exhibit a high degree of coexpression and likely generate a regulon. Inactivation of any PAP gene results in an albino or ivory phenotype accompanied by elevated NEP activity and increased expression of NEP-dependent chloroplast genes [4]. When exposed to light, *pap* mutants develop a normal photomorphogenetic phenotype and plastids without thylakoid formation [5]. A block of chloroplast biogenesis generates severe stress induced via dysregulation of the plant defense system and has a broad impact on hormone signaling circuits [6]. Therefore, *PAP*s may be an exclusive tool for dissecting interactions between chloroplast biogenesis and phytohormone pathways. Their interplay, in turn, can modulate the expression of nuclear genes encoding proteins with non-plastidial locations.

The role of phytohormones in this mutual exchange of information between chloroplasts and the nucleus is far from understood. A detailed examination has revealed the differential effects of exogenously applied phytohormones on the transcript levels of genes related to the plastid transcription machinery [7]. Unlike CKs and IAA, which have stimulatory effects, ABA, methyl jasmonate, and SA repress the transcript accumulation of *PAP* genes in the wild type, whereas no reproducible alterations are observed with gibberellic acid, brassinolide or the ethylene precursor 1-aminocyclopropane-1-carboxylic acid. It should be noted, however, that the transcriptional responses to exogenous hormone applications depend on age and developmental stage and may be near saturation in wild-type (WT) plants.

Gene expression profiling of the *pap7* mutant recently presented by Grübler et al. [6] revealed several hormone-related genes regulated by biogenic retrograde signals from plastids. Among them, two highly downregulated genes, *LOX2* (*lAt3G45140*) and *AOS* (*At5G42650*), encode lipoxygenase and allene oxide synthase, respectively, key enzymes involved in the production of oxophytodienoic acid (OPDA), a precursor of peroxisomal jasmonic acid. On the other hand, the expression of *ACO3* (*AT1g12010*) and *CORI3* (*AT4G23600*), involved in ethylene biosynthesis, was upregulated and downregulated, respectively, indicating that disruption of *PAP* genes may have both negative and positive effects on hormone-related genes.

Recently, we have shown that impaired expression of the *PAP1* and *PAP6* genes in *Arabidopsis* contributed to the transition of CK-dependent accumulation of chloroplast gene transcripts and transcripts of the genes for plastid transcription machinery from positive to negative, although the CK signaling in the mutants remained practically unperturbed [8]. The opposite response was due to hormone overproduction as a consequence of the constitutive overexpression of the genes involved in CK synthesis and degradation. Elevated CK content could be the result of a compensatory mechanism allowing plants to promote morphogenesis in the absence of regular photosynthesis.

Extensive crosstalk and signal integration among growth-regulating hormones imply that CK overproduction may affect the transcript levels of genes involved in the metabolism of other hormones. A complex network of such interactions suggests that long-term effects of the altered content of any hormone can induce a “domino effect”, resetting many systems within the plant [9]. Compensatory processes aimed at maintaining hormone homeostasis in *pap* mutants are supposed to launch rearrangements of transcriptional programs to ensure plant viability under highly modulated conditions. At least in part, these differences can be interpreted as a consequence of retrograde signaling from defective plastids.

In this work, we applied highly sensitive analysis of phytohormone content and qRT–PCR to examine the effect of *pap* mutations on plant hormonal status and possible implications for the expression of chloroplast and chloroplast-related nuclear genes.

## 2. Results

### 2.1. Disruption of Pap Genes Contributes to Altered Hormone Levels

We employed two mutants with disrupted *PAP* genes, *pap1*/*ptac3* (NASC 639260) and *pap6*/*fln1* (NASC 305140), which exhibit features typical of so-called “PAP syndrome” [5]. Both mutants had albino phenotypes (Figure 1A,B) with reduced PEP-dependent plastid transcription and were able to grow only in the presence of a carbon source. PAP1 protein belongs to a group implicated in DNA/RNA metabolism. It contains a SAP DNA-binding domain and a PPR motif and has been shown to bind in a light-dependent manner to the transcribed regions of all PEP-dependent photosynthesis and rRNA genes [10]. The fructokinase-like protein PAP6/FNL2 is a member of a functional group engaged in redox regulation. PAP6/FLN2 is a target of PAP10/TrxZ, a redox-active thioredoxin, with which it interacts via cysteine bonds to form disulfide bonds [11,12]. However, complementation analysis suggested that PAP6 and PAP10 are structural rather than functional components of the PEP complex [3].

Previous studies have indicated that disruption of *PAP* genes results in elevated levels of cytokinins and their precursors [8]. In contrast, the levels of the CK antagonist abscisic acid (ABA) were significantly reduced as shown by our experiments. Both mutants accumulated 2 times less ABA than did the wild type and had greatly decreased levels of ABA metabolites, namely, phaseic acid (PA), 9-hydroxy-ABA (9OH-ABA) and dihydrophaseic acid (DPA), in *pap6* (Table 1). Similar trends, even more highly enhanced, were observed for the levels of ACC, a precursor of ethylene in *pap1*. The mutants contained no more than 3–10% of the wild-type values.

On the other hand, the levels of benzoic acid (BzA) and notably of salicylic acid (SA) significantly increased. The *pap1* mutant accumulated 3 times more SA, and the *pap6* mutant accumulated 4 times more SA than did the wild type, which indicates the possible activation of SA biosynthesis. There were no significant differences in the levels of jasmonic acid (JA) or its bioactive metabolite JA-isoleucine (JA-Ile) between the wild type and the mutants. In addition, the content of the JA precursor cis-oxophytodienoic acid (cisOPDA) decreased, especially in *pap6.*

Analysis of the *pap* mutants revealed no differences in free IAA levels when compared with those of the wild type; however, the amounts of the IAA precursor indole-3-acetonitrile (IAN) and IAA metabolites, including IAA-glucose ester (IAA-GE), oxo-IAA and oxo-IAA-glucose ester (OxIAA-GE), tended to increase, albeit not always significantly. The levels of naturally occurring phenylacetic acid (PAA) also increased in the mutants. However, the biological significance of PAA is not completely understood. In addition to its auxin activity, PAA has been hypothesized to regulate the effects of free IAA by inhibiting polar auxin transport [13]. Notably, the content of IAA-aspartate (IAA-Asp) was significantly lower in both *pap1* and *pap6* than in the WT. A decrease in IAA-Asp content in *pap* mutants suggested a bias toward enhanced accumulation of auxin storage forms, which are postulated to regulate auxin homeostasis in growth and development [14].

### 2.2. The Expression of Genes Involved in Hormone Metabolism Is Altered in Pap Mutants

Since the levels of various hormones are altered in *pap* mutants compared to those in WT plants, it is conceivable that the expression of at least some of the genes involved in their synthesis and metabolism was altered in these mutants. However, the search for the particular genes responsible for the altered hormonal status of *pap* mutants is impeded by the complexity of the metabolism of some phytohormones coupled with the specific regulation of different members of hormone gene families. To determine whether the transcript abundance of hormone-related genes and their response to treatment with exogenous phytohormones are associated with the levels of the corresponding hormones, we examined the expression of the key biosynthesis genes. Our studies were based on generally accepted concepts of hormone biosynthetic and signaling pathways and, in part, on publicly available transcriptomic data for *pap7*, taking into account the coregulation paradigm of *PAP* genes [6].

#### 2.2.1. JA and SA

Consistent with the previously reported data for *pap7*, a block of chloroplast biogenesis had a strong impact on two genes encoding key enzymes for OPDA production, *LOX2* and *AOS.* We performed a qRT-PCR assay to determine whether these two genes exhibited the same expression patterns in the *pap1* and *pap7* mutants. According to our analysis, the expression of both genes was downregulated 3–5-fold compared to that in the WT (Figure 2A). In addition, steady-state levels of the JA marker gene *Thi2.1* (*At1g72260*) and the JA and ET marker gene *PDF1.2* (*At5G44420*), which encode the antimicrobial proteins thionin and defensin, respectively, strongly decreased in both the mutants and in *pap7*, as shown from the transcriptomic data.

OPDA production is tightly regulated by SA, which was shown to decrease the expression of the genes encoding the OPDA biosynthetic enzymes LOX2, AOC2 and AOS in Arabidopsis [15]. OPDA levels can in turn affect the transcriptional activity of SA-related genes. In *pap* mutants, the decrease in the expression of OPDA synthesis genes was associated with the absence of thylakoid lipids, precursors of allene oxide [6]; therefore, jasmonate appears to act upstream of salicylate (Figure 2B). As can be judged from the results of our experiments, elevated levels of SA in *pap* mutants corresponded to increased expression of the SA biosynthesis-related gene *ICS1 *(isorchorismate synthase 1), which encodes a crucial enzyme in the SA synthesis pathway that accounts for 90% of SA production [16]. Elevated transcript levels of *ICS1* were accompanied by increased expression of the SA signaling gene *NPR1* (non-expressor of pathogenesis-related genes1), a master regulator of downstream SA signaling [17], and the SA marker gene *PR1* (PATHOGENESIS-RELATED 1) [18]. These data support the antagonistic relationships between SA and OPDA, which are involved in plant defense responses.

#### 2.2.2. IAA

Chronic SA overproduction can be associated with a concomitant reduction in auxin biosynthesis, transport and signaling [19]. According to the transcriptomic data, several auxin-responsive genes, *AT1G29430*, *AT4G12980*, and *AT1G34310* (*AXLIN RESPONSE FACTOR 12* and *SAUR21*), were repressed in *pap7*, with an expression change exceeding the threshold of 1 (log_2_). Downregulation of *SAUR21*, which encodes SMALL AUXIN UP RNA, was also confirmed in our experiments via qRT-PCR for *pap1* and *6* (Figure 2C). In addition, it has been reported that SA repression of auxin-related genes has no significant effect on free auxin levels [20]. This finding is consistent with the almost identical content of bioactive IAA in the WT and pap mutant strains detected in our studies (Table 1). In accord with this, the expression of the *TAA* and *YUCCA* genes, which encode major steps of IAA biosynthesis from tryptophan via the IPA (indole-3-pyruvic acid) pathway [21], did not significantly differ between *pap7* and the WT.

It should be noted, however, that only a small fraction of auxin exists as free, active signaling molecules [13]. Therefore, altered levels of auxin precursors and conjugated forms may contribute to the maintenance of auxin homeostasis in *pap* mutants under conditions of arrested chloroplast biogenesis. In this regard, the decreased expression of *SUR1* (*SUPER ROOT1*) is of interest. *SUR1* encodes the enzyme involved in the conversion of the IAA precursor IAOx to indole glucosinolate (IGS) in the side branch of the IAOx pathway [21]. Inactivation of *SUR1* disrupts IGS biosynthesis and promotes accumulation of the upstream intermediate IAOx, which was elevated in *pap1* and *6* (Table 1).

In addition, the expression of *Gretchen*
*Hagen 3* (*GH3.9*), a member of the family of genes encoding amido synthases, was strongly downregulated in the *pap* mutants, which was in line with the significantly reduced levels of IAA-Asp (Figure 2C; Table 1). Amido synthases conjugate amino acids to carboxyl groups of small molecules, including auxin [22]. However, the dependence of this repression solely on auxin is ambiguous since the reduced expression of *GH3.9* may also be related to SA or JA metabolism. Therefore, the influence of IAA production and its effects are multifaceted and aim to balance plant growth and defense systems through cross interactions with other plant hormones.

#### 2.2.3. Ethylene and ABA

Next, we analyzed the expression of ethylene biosynthesis genes. Ethylene is produced from its general precursor S-adenosyl-L-methionine (SAM) via a two-step biosynthesis route. In the first step, SAM is converted to ACC and 5′-methylthioadenosine (MTA) by ACC synthase (ACS). In the second step, ACC is converted to ethylene, CO_2_ and cyanide by ACC oxidase [23]. The *ACSs* constitute a multigene family that is differentially expressed in plants. The transcript accumulation of two members of the *ACS* family, *ACS4* and *ACS8*, was highly repressed in the *pap1* and *pap6* mutants, consistent with the drastically reduced ACC content (Figure 2D; Table 1). Both genes also displayed low expression levels in *pap7* (−1.0 log_2_-fold for *ACS4* and −2.1 for *ACS*8) [6]. Along with these findings, the accumulation of *ACO3* transcripts in *pap7*, as well as in *pap1* and *pap6*, was twice as high as that in the wild type. This result may reflect the specific induction of *ACO3* in response to a decrease in the level of ACC.

Among the multiple members of the ethylene response factor (ERF) gene family, *ERF1* and *ERF14* had more than a 50% decrease in the magnitude of change in expression in the *pap* mutants compared to the WT (Figure 2D). These two *trans*-factors are of particular interest because they were shown to function in ethylene/jasmonic acid plant defense responses [24]. In addition, ERF1 upregulated specific suites of genes in response to different abiotic stresses [25]. Since both ERF1 and ERF14 are key elements in the integration of ethylene and JA signals, their suppression in *pap* mutants, as well as impairment of ACC synthesis, may also be a consequence of a decrease in OPDA production.

The ethylene biosynthesis genes *ACS4* and *ACS8* were shown to be negatively regulated in *Arabidopsis* by ABA through ABI4-mediated transcriptional repression [26]. The expression of *ABI4*, as well as that of the ABA marker gene *RD29*, decreased in the *pap* mutants (Figure 2E). In parallel, among the genes of the NCED family that control the entire process of ABA biosynthesis by regulating the rate-limiting step [27], *NCED4* was significantly downregulated in *pap7* (−1.66 log_2_) [6]. According to our qRT-PCR assays, in addition to *NCED3* and *ABA2*, *NCED4* was also strongly repressed. A decrease in *NCED* expression as well as upregulation of the ABA catabolism gene *CYP707A*1 could account for the reduced ABA levels, thus demonstrating a positive feedback interaction between ethylene, OPDA and ABA in the mutants.

In summary, arrest of plastid development induces reprogramming of the expression of hormone biosynthetic and metabolic genes in *pap* mutants, which in turn leads to concomitant defects in the content and likely interplay between various hormones.

### 2.3. Pap Mutants Exhibited Partially Altered Responses to Hormone Treatments

Next, we analyzed whether the modified hormone status of *pap* mutants affects hormone signaling and response to hormone treatment. As previously indicated, the expression levels of the ABA marker *RD29*, the ethylene and JA marker *PDF1*,*2* and the IAA marker *IAA19* were lower in the *pap* mutants than in the WT plants. The relative induction of these genes exceeded that of the WT plants, indicating increased responsiveness of the mutants to exogeneous hormones (Figure 3A). In contrast, the expression of the SA marker gene *PR1* was downregulated following the application of exogenous hormones in accordance with the activation of a positive feedback loop, diminishing the uncontrolled activation of hormone signaling. Overall, these results indicate that hormone signaling is still functional in *pap* mutants.

Several bioassays were used to assess the sensitivity of the strains to ACC, ABA, IAA, metJA and SA. For shoot elongation tests, the seedlings were grown for 6 days in the dark in media supplemented with a range of hormone concentrations. Homozygous *pap1* and *pap6* hypocotyls were identified based on the white phenotype of the cotyledons after they had been cultivated for an additional 24 h under light. The hypocotyl lengths of the mutants did not significantly change relative to those of the WT plants after treatment with IAA, SA, ACC or metJA (Appendix A). Therefore, at the early stages of development, the plants responded to hormonal treatment just as the wild-type plants did. This finding is consistent with the observation that these mutations do not affect the skotomorphogenic program and that *pap* mutants develop normally within the first few days even without sugar [5].

Compared with those of Col-0, the roots of the *pap1* and *pap6* plants were significantly shorter (41.9 ± 13.3% and 43.8 ± 6.6%, respectively) (Figure 1B). In the presence of SA, ABA and IAA, the relative decreases in lengths were similar to those in the wild-type plants. However, when grown on plates containing various concentrations of ACC or metJA, both mutants exhibited considerably shortened roots compared with corresponding decreases in Col^−0^ (Figure 3B,C; Appendix A). From these results, we conclude that the mutants were more sensitive to treatment with these hormones.

Since the mutants were deprived of chlorophyll, the ability of the tested hormones to retard or accelerate senescence in detached leaves was evaluated based on changes in protein content. With SA and IAA treatments, protein degradation did not significantly differ from that in water-treated samples for either the wild type or the mutants (Figure 3C, Appendix A). ABA, ACC and metJA contributed to an accelerated decrease in protein content, and the decrease was significantly greater in mutants following the application of metJA and ACC. Therefore, *pap* mutations affect the ability of ACC and metJA to accelerate leaf senescence.

It thus appears that hormones may play specific roles in various stages of development in *pap* mutants, participating in some aspects of developmental processes. At each stage, the physiological relevance of the altered sensitivity to hormones should be considered separately.

Our previous studies revealed the important regulatory role of plant phytohormones in regulating the expression of genes related to the chloroplast transcription machinery [7,28]. However, *pap* mutations did not substantially modify the response to hormone application. The transcript accumulation of *NEP*, *SIG1-4*,*6* and operational *PAP* genes was upregulated by IAA but did not respond to or was slightly activated by ACC treatment (Figure 4, Appendix A). The genes were repressed by metJA and in part by ABA. The exceptions were *CKA4*, which encodes the Ser/Thr protein kinase cPCK2, and stress-responsive *SIG5*, which were upregulated by ABA in both the WT and mutant plants, consistent with the upregulation of these genes by ABA shown by Wang et al. [29] and Yamburenko et al. [30]. In parallel, the latter two genes were downregulated by IAA, exhibiting a special type of hormonal regulation. SA treatment did not affect the expression of the genes involved in the chloroplast transcription machinery or contributed to their slight suppression.

Considering the highly reduced expression of genes with PEP promoters (class I) and their residual transcript levels, our studies were mainly focused on modulations of the expression of selective NEP-dependent (class III) and NEP- and PEP-dependent (class II) genes. The expression of the studied chloroplast-encoded genes was gene-specific and in part divergent from that of the WT plants. In particular, under our experimental design, *rpoB* (the beta subunit of PEP) was significantly upregulated by IAA, while *clpP* (an ATP-dependent protease) and *accD* (an acetyl-CoA carboxylase) were repressed by metJA and ABA only in the mutants; these genes did not respond reliably to hormone treatment in the WT plants. These data confirmed the increased sensitivity of the mutants to IAA, metJA and ABA. In both the WT and mutant strains, no reproducible changes were observed when the samples were exposed to SA or ACC treatment.

## 3. Discussion

Mutants with functionally inactivated PAPs provide unique tools for the dissection of hormone response networks in plants. The dramatic fluctuations in the contents of major phytohormones detected in our studies suggest that changes in hormone interplay occur mainly at the level of hormone metabolism, while signaling circuits mostly remain operational. The most marked differences between the *pap* mutants and the wild type relate to the greatly increased SA content and the concomitant decreases in ACC and OPDA, which are precursors of ethylene and jasmonic acid, respectively, as well as a significant reduction in ABA levels (Figure 5). In addition, we previously detected CK overproduction as a result of constitutive overexpression of genes involved in the synthesis and degradation of CK [8].

The magnitude and direction of the changes in expression of hormone metabolism genes suggest that they are triggered by altered sink/source relationships induced by nonautotrophic metabolism in albino plants. A recent transcriptomic study of *pap7* revealed that a block in chloroplast biogenesis had a significant impact on metabolic genes related to starvation processes and the mobilization of storage energy [6]. Thus, the transcriptomic data revealed the induction of several genes encoding proteins involved in sucrose degradation and transport and variably regulated fatty acid metabolism. In particular, a lack of linoleic acid in arrested plastids contributed to the downregulation of the allene oxide pathway and the production of OPDA, the precursor of jasmonic acid, which is highly important for the pathogen defense system. These results are also consistent with microarray data from the Arabidopsis *immutans* (*im*) mutant, which has green and white sectoring due to the action of the nuclear recessive gene *IMMUTANS* (*IM*). The expression of genes encoding key enzymes of the allene oxide pathway (AOS, AOC1, AOC4 and LOX2) was strongly repressed in the white sectors of the mutant (4–10-fold) compared to that in the WT leaves [31].

Severe stress generated by arrested plastids activates different protective systems in response to the highly altered metabolism of white tissues. SA, the production of which is strongly increased in *pap* mutants, is known to play a central role in local and systemic acquired resistance against biotrophic pathogens, while the JA-mediated response coupled with the ethylene signaling pathway contributes to defense against necrotrophic pathogens [17]. The SA- and ethylene/JA-mediated defense pathways are mutually antagonistic, which likely accounts for the elevated SA levels in the mutants with reduced OPDA and ACC contents. Hence, we speculate that SA accumulation can be treated as a response to enhanced susceptibility to biotrophic pathogens such as *Pseudomonas syringae.* Similarly, increased vulnerability has been detected for the white sectors of the *im* mutant with reduced lignin and cellulose microfibrils, and alterations in galactomannans and the decoration of xyloglucan [32].

However, the involvement of SA in the regulation of genes associated with pathogenesis may not be the only factor determining its increased content. Recent evidence suggests that SA can act antagonistically to the inhibition of plastid biogenesis by promoting the accumulation of photosynthesis-associated proteins [33]. Exogenous SA partially restored the level of chlorophyll in norflurazon (NF)-treated Arabidopsis plants and in the *plastid protein import2* (*ppi2*) mutant, which had reduced SA and JA contents and an albino phenotype, and in addition, it increased the levels of some photosynthesis-associated proteins. In accordance with the positive effect of SA on chloroplast biogenesis, light regulation of *PhANGs* (with the exception of *LHCB* genes) and other nuclear gene groups was fully functional in the *pap7* mutant, indicating that a block in chloroplast biogenesis does not repress their expression [5]. Conversely, SA is known to promote senescence and decrease levels of chlorophyll [34]. These findings suggest that SA may play opposite roles in the regulation of chlorophyll content, which are hypothesized to be determined by specific signaling pathways activated by SA [33].

The role of SA in balancing stress responses and growth can be at least partially realized via interactions with auxin and CK. SA attenuates plant growth by regulating the biosynthesis, transport and signaling of auxin [19], which is consistent with the downregulation of several IAA-related genes observed in our studies (*SAUR21*, *GH3.9* and several auxin-responsive genes). On the other hand, elevated SA levels in *pap* mutants were superimposed on increased levels of certain auxin precursors and conjugated forms, as well as concomitant enhanced accumulation of CK metabolites and precursors [8]. The ability of CK to increase *ICS1* expression was shown by Choi et al. [35]. In response to non-cytokinin-secreting pathogens, *ICS1* was hyperactivated in the presence of CKs. Hence, SA-mediated responses are multilayered and involve both the reinforcement of defense mechanisms and the activation of plant growth through crosstalk with other plant hormones, such as auxin and CK.

An elevated content of CK precursors and metabolites in *pap* mutants may augment sink activity and promote morphogenesis in the absence of normal photosynthesis. The ability of CK to create new source-sink relationships and increase the nutrient sink activity of plant tissues to support their growth was first described by Mothes [36,37]. We therefore hypothesize that alterations in the levels of growth-promoting hormones in *pap* mutants may be the result of sink demand in non-photosynthesizing tissues to optimize plant growth.

Along with the upregulation of genes encoding phytohormones with growth-promoting activity, genes encoding hormones with inhibitory effects were largely suppressed. This group, which includes ethylene-, ABA- and JA-related genes, triggers phenotypic changes in response to biotic and abiotic stresses under a variety of specific environmental conditions. The downregulation of these genes and the parallel decrease in hormone content imply that uncontrolled activities of the aforementioned hormones may contribute to severe physiological perturbations in *pap* mutants and failure to vegetate even on Suc-supplemented media. This finding is consistent with the overall trend suggesting that crosstalk between growth-promoting and growth-repressing hormones often opposes each other.

The specific feedback loops regulating the activity of various hormones in *pap* mutants can be attributed to the impact of retrograde signals from arrested albino plastids. Plastidial control of nuclear-encoded hormone-associated genes is partly explained by the fact that plastids are the sites of synthesis of a number of hormones, including CK, ABA, JA and SA. Most of the genes identified in the “Hormone” subset of the *pap7* mutant were shown to encode proteins with non-plastidial locations, which was ascribed to the broad impact of retrograde biogenic signals on the hormone signaling network [6].

Changes in the hormonal status of *pap* mutants can remodel the responses of plastid-encoded genes to hormone treatment from positive to negative and vice versa. Thus, the disruption of *PAP* genes contributed to the abolishment of the positive CK effect on the accumulation of chloroplast gene transcripts and transcripts of the genes for plastid transcription machinery [8]. However, SA treatment did not affect the accumulation of plastid gene transcripts, probably due to the optimal concentration of SA, whereas at least some chloroplast genes in *pap* mutants were found to be more sensitive to metJA and ABA, consistent with enhanced signaling responses to these hormones.

## 4. Materials and Methods

### 4.1. Plant Material, Growth Conditions and Hormone Treatment

*Arabidopsis thaliana* ecotype Columbia 0 was used in all the experiments. The *pap1* (NASK ID N 639260) and *pap6* (NASK ID N305140-45) mutants were obtained from the European Arabidopsis Stalk Center. Seeds were sown on MS media supplemented with 2% sucrose and grown in a controlled-environment chamber under a 16-h light and 8-h dark photoperiod at an illumination of 100 μmol m^−2^ s^−1^ and a temperature of 23 °C for 4 weeks. For hormone treatment, plants were sprayed with a solution of hormones or an equal aliquot of ethanol and collected after 3 h of exposure. The compounds assayed were abscisic acid (ABA, 5 × 10^−5^ M), indole-3-acetic acid (IAA; auxin, 10^−6^ M), 1-aminocyclopropane-1-carboxylic acid (ACC; ethylene precursor, 10^−5^ M), salicylic acid (SA, 10^−5^ M) and methyl jasmonate (MetJA 5 × 10^−5^ M). The concentrations of the active reagents and treatment times were selected based on preliminary experiments. Leaves from at least six different plants were harvested for each replicate to average the results, after which the probes were stored at −80 °C. All the experiments were performed in at least three biological replicates.

### 4.2. Hormone Extraction, Purification and Determination

Frozen samples (100 mg FW) were homogenized and extracted with cold (–20 °C) methanol/water/formic acid (15/4/1, *v*/*v*/*v*) as described in Dobrev and Kaminek [38] and Dobrev and Vankova [39]. Phytohormones were separated with a reverse-phase–cation exchange SPE column (Watters Oasis-MCX; WICOM Germany GmbH, Heppenheim, Germany) into the acid fraction by elution with methanol and into the basic fraction by elution with 0.35 M NH_4_OH in 60% methanol, which were used for the determination of auxins, ABA, SA, jasmonates and ACC. Data processing was performed with the Analyst 1.5 software package (Applied Biosystems, Waltham, MA, USA).

### 4.3. Quantitative RT-PCR

Total RNA was isolated from 30-day-old rosette leaves using the TRIzol (Thermo Fisher Scientific, Waltham, MA, USA) method. RT-PCR was performed using a LightCycler 96 (Roche, Basel, Switzerland) with hot-start SYBR Green I technology and gene-specific primers according to the protocol described previously [40]. The primers used are listed in the Appendix A. The following standard thermal profile was used for all PCRs: 95 °C for 5 min; 40 cycles of 95 °C for 15 s, 58 °C for 15 s and 72 °C for 25 s. *UBQ*10 (*at5g53300*) and protein phosphatase 2A (PP2A) regulatory subunit A2 (*at3g25800*) were used as internal controls. The primers used are presented in Appendix A.

### 4.4. Growth Sensitivity Assays

For the hypocotyl elongation assay, seeds were grown in darkness for 4 days on MS media supplemented with various concentrations of phytohormones. Homozygous *pap1* and *pap6* plants were identified based on the white phenotype of the cotyledons after they had been cultivated for additional 24 h under light.

For the root elongation assay, 7-day-old seedlings were subsequently transferred to MS plates supplemented with various concentrations of hormones and grown vertically for another 4 days under 16 h light/8 h dark conditions. Measurements were performed in triplicate with 20 seedlings for each experiment.

A protein retention assay was performed with rosette leaves excised from the 3rd and 4th layers of 5-week-old plants. The leaves were placed on filters moistened with hormone solutions or water with a solvent and kept for 3 days in darkness. Total soluble protein concentration was determined by the bicinchoninic acid (BCA) method using a Pierce BCA assay kit (Thermo Fisher Scientific, Waltham, MA, USA) and is expressed as µg/mm^2^. The protein content at the start of the experiment was taken as a reference and set at 100%.

### 4.5. Statistical Data Processing

The experiments were performed in triplicate. The significance of differences was estimated with one-way analysis of variance (ANOVA) followed by Tukey’s method using an online calculator (astatsa.com/OneWayANOVA_with_TukeyHSD/) (accessed on 10 December 2023) and Student’s *t* test. All the data are presented as the means ± standard errors (SEs).

## 5. Conclusions

Mutants of PEP-associated proteins provide an excellent opportunity to expand our understanding of possible interactions among phytohormones and clarify the mechanisms of retrograde signaling from in defective plastids. The block of chloroplast biogenesis and the ability to vegetate only in the presence of an external carbon source cause reprogramming of the expression of hormone-related genes and specific alterations in hormonal status, which play important roles in plant growth strategies. Elevated levels of SA, which balance stress responses and plant growth, are combined in *pap* mutants with increased levels of certain auxin precursors and conjugated forms, as well as a concomitant increase in the accumulation of CK metabolites and precursors that collectively promote morphogenesis in the absence of normal photosynthesis. Conversely, the levels of JA, ethylene and ABA, which negatively regulate morphogenesis, were significantly lower in the transgenic plants than in the WT plants. Hence, the interplay between growth-promoting and growth-suppressing hormones remains opposite in *pap* mutants, despite specific multilevel changes in hormone metabolic pathways. *Pap* mutations do not substantially modify the responses of chloroplast genes to hormone application, although in some cases, the effects are gene specific and are selectively divergent from those in green plants.

Taken together, our results indicate that the transition from autotrophic to heterotrophic metabolism in *pap* mutants induces a concerted transcriptomic response promoting versatile shifts in hormone metabolic pathways. Even though the exact set of specific transcriptional targets of these hormone-related changes still needs to be determined, the fact that altered hormone status can ensure the viability of nonphotosynthetic plants under modulated conditions is intriguing and suggests bifurcated pathways of hormone-related growth regulation.

## Figures and Tables

**Figure 1 ijms-25-01586-f001:**
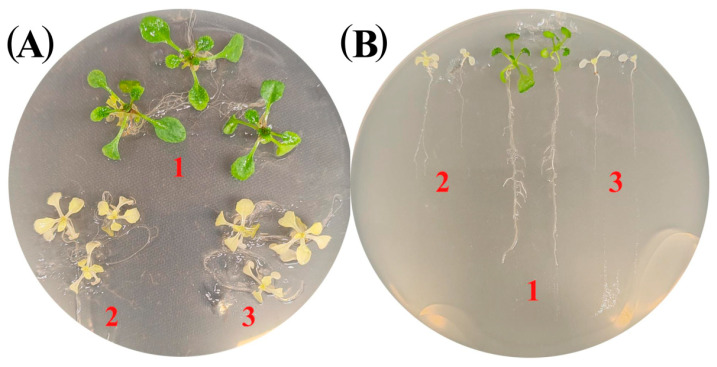
(**A**) Phenotypes of *pap* mutants and wild-type plants grown on Murashige and Skoog (MS) media in Petri dishes for 4 weeks (**A**) under an illumination of 100 μmol m^−2^s^−1^ and a temperature of 23 °C with a 16 h photoperiod. 1-WT: Col-0, 2*-pap1*, 3-*pap6*. (**B**) Root elongation assay. Seven-day-old seedlings were transferred to vertical MS plates and grown vertically for another 4 days under 16 h light/8 h dark conditions: 1, Col-0; 2, *pap1*; and 3, *pap6*.

**Figure 2 ijms-25-01586-f002:**
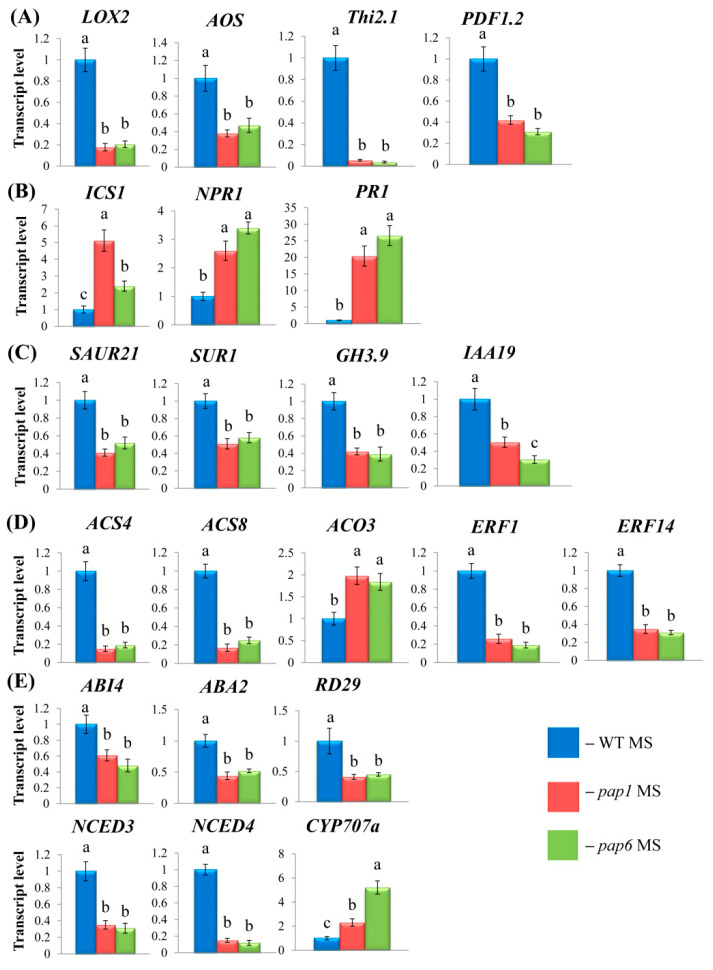
Relative expression values of *A. thaliana* hormone biosynthesis and signaling genes in untreated 4-week-old wild-type plants and *pap* mutants: (**A**) JA, (**B**) SA, (**C**) IAA, (**D**) ethylene, (**E**) ABA. WT and mutant plants were grown on MS media in Petri dishes for four weeks under a 16 h light/8 h dark photoperiod at 23 °C with 100 μmol m^−2^s^−1^. Total RNA was isolated from rosette leaves and analyzed via qRT-PCR using *UBQ10* and *PP2A* as internal standards. The data presented in the figure are the mean values (n ≥ 3). Error bars represent SEs. Different letters denote statistically significant differences at *p* < 0.05 (ANOVA with post hoc Tukey’s multiple-comparison test).

**Figure 3 ijms-25-01586-f003:**
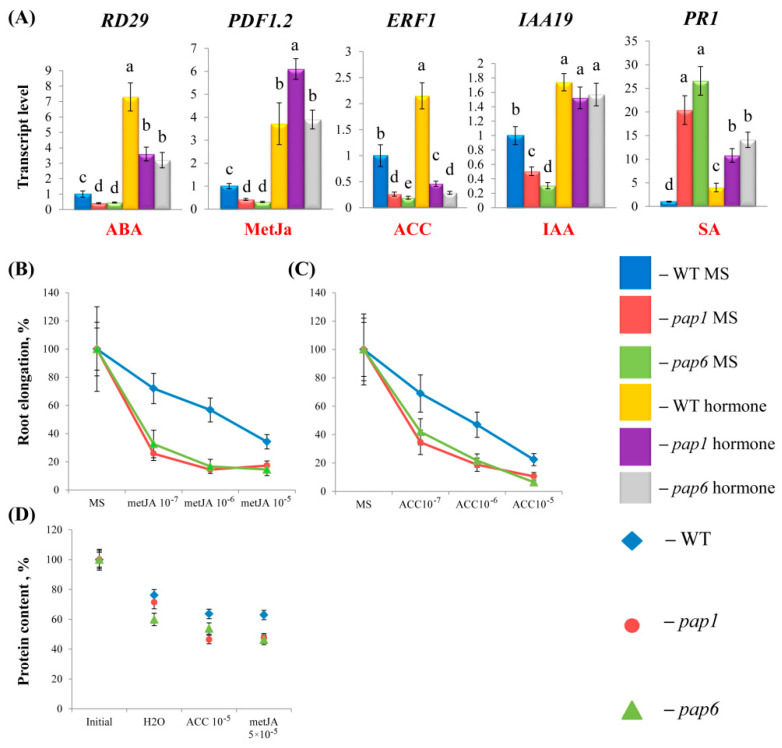
(**A**) Induction of marker genes reflecting the activation of different hormonal pathways in 4-week-old wild-type plants and *pap* mutants. WT and mutant plants were grown on MS media in Petri dishes for four weeks under a 16 h light/8 h dark photoperiod at 23 °C with 100 μmol m^−2^s^−1^ and treated with solutions of hormones or an equal aliquot of ethanol for 3 h. Total RNA was isolated from rosette leaves and analyzed via relative quantitative RT-PCR using *UBQ*10 and *PP*2*A* as internal standards. The data are presented as the means (n ≥ 3). Error bars represent SEs. Different letters denote statistically significant differences at *p* < 0.05 (ANOVA with post hoc Tukey’s multiple-comparison test). (**B**,**C**). Primary root growth inhibition in response to exogenous hormones. Seedlings were grown on MS media for 7 days under 16 h light/8 h dark conditions, subsequently transferred to vertical media plates supplemented with a range of concentrations of metJA and ACC and grown for another 4 days. Measurements were performed in triplicate with 20 seedlings for each experiment. Error bars represent SEs. (**D**) Sensitivity to hormone treatment during dark-induced senescence. Total protein content was measured in detached leaves (the 3rd and 4th layers) of *A. thaliana* WT plants and *pap* mutants incubated in the dark at 23 °C on water or metJA (5 × 10^−5^ M) and ACC (10^−5^ M) for 3 days.

**Figure 4 ijms-25-01586-f004:**
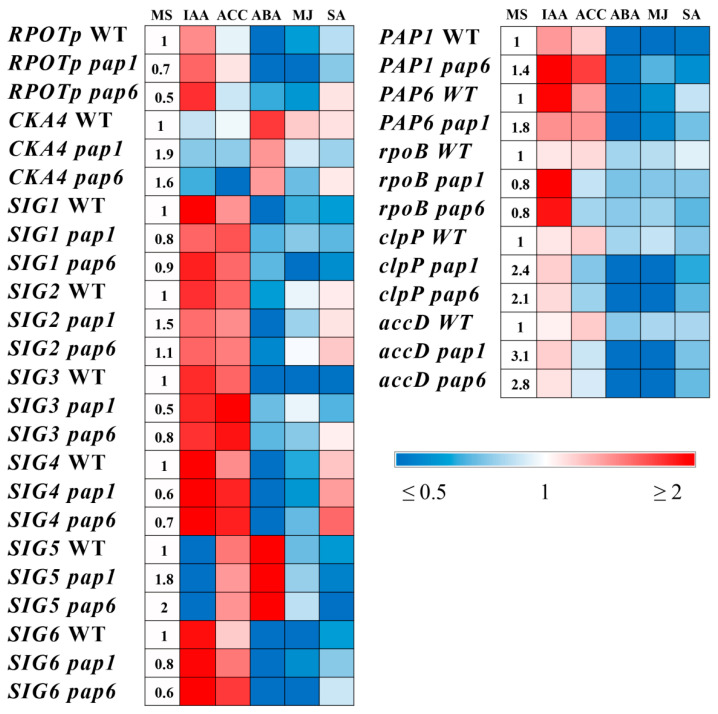
Effect of hormone treatment on the relative expression values of genes related to the chloroplast transcription machinery and chloroplast-encoded genes. WT and mutant plants were grown on MS media in Petri dishes for four weeks under a 16 h light/8 h dark photoperiod at 23 °C with 100 μmol m^−2^ s^−1^ and treated with solutions of hormones or an equal aliquot of ethanol for 3 h. Total RNA was analyzed via quantitative RT-PCR using *UBQ*10 and *PP*2*A* as internal standards. The data are presented as the means (n ≥ 3). The numbers in the “MS” column indicate the baseline ratio of the expression of each gene in the wild type and the mutants without treatments. All the numerical data are presented in Appendix A.

**Figure 5 ijms-25-01586-f005:**
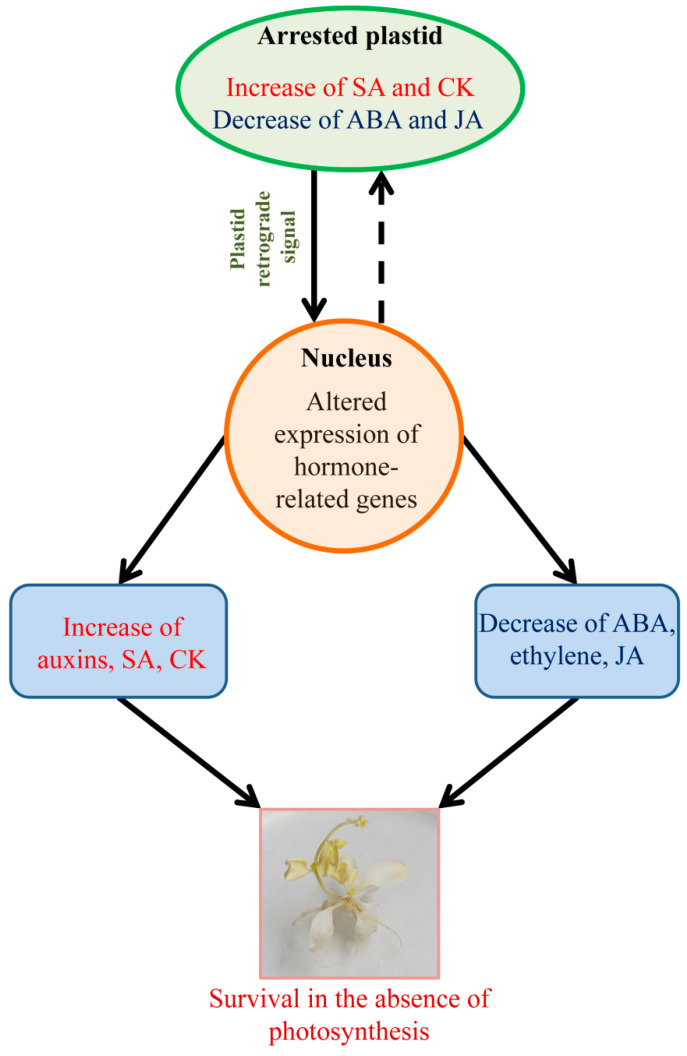
Schematic representation of changes in hormone content of *pap* mutants. Plastid signals from arrested plastids alter the expression of hormone-related genes, resulting in an increase of salicylic acid, cytokinins and auxins and a decrease of abscisic acid, ethylene and jasmonates. Data on cytokinin content are based on the results given in [8].

**Table 1 ijms-25-01586-t001:** The impact of *pap* mutations on the phytohormone content (pmol/g fresh weight) of *A. thaliana* rosette leaves.

Phytohormones	WT	*pap1*	*pap6*
ABA	358.91 ± 72.24	178.43 ± 9.84 *	168.73 ± 20.68 *
PA	84.39 ± 12.01	14.64 ± 2.62 **	6.73 ± 2.16 **
9OH-ABA	15.01 ± 2.8546	2.70 ± 0.46 **	1.14 ± 1.14 **
DPA	31.22 ± 7.04	29.49 ± 7.02	18.76 ± 3.91
BzA	655.04 ± 61.87	1092.02 ± 146.66 *	1160.01 ± 90.71 *
SA	882.35 ± 182.37	2588.49. ± 498.01	3264.72 ± 897.05 *
JA	151.58 ± 29.71	145.89 ± 30.01	140.03 ± 13.77
JA-Ileu	16.31 ± 3.27	9.3225 ± 3.03	14.17 ± 3.50
cisOPDA	137.52 ± 7.42	94.85 ± 24.90	58.77 ± 13.50 *
ACC	338.17 ± 86.45	11.00 ± 2.63 **	34.28 ± 7.05 **
IAA	67.44 ± 2.89	64.63 ± 6.54	66.74 ± 4.22
OxIAA-GE	922.21 ± 129.18	1461.79 ± 151.29 *	1232.99 ± 119.36
PAA	98.06 ± 8.44	133.42 ± 21.69	162.57 ± 18.38
IAA-Asp	56.51 ± 17.94	5.80 ± 1.12 *	12.76 ± 2.35 *
IAA-GE	20.11 ± 1.91	33.34 ± 5.95	32.02 ± 5.82
OxIAA	248.51 ± 38.26	312.15 ± 47.86	326.04 ± 26.21
IAN	4284.60 ± 1411.27	5577.52 ± 833.42	6208.17 ± 2741.61

Abbreviations: ABA—abscisic acid, PA—phaseic acid, DPA—dihydrophaseic acid, 9OH-ABA—9-hydroxy-ABA, BzA—benzoic acid, SA—salicylic acid, JA—jasmonic acid, JA-Ile—JA-isoleucine, cisOPDA—cis-oxophytodienoic acid, ACC—1-aminocyclopropane-1-carboxylic acid, IAA—indole-3-acetic acid, IAA-GE—IAA-glucose ester, PAA—Phenylacetic acid, IAA-Asp—IAA-aspartate, OxIAA—oxo-IAA, OxIAA-GE—oxo-IAA-glucose ester, IAN—Indole-3-acetonitrile. Asterisks (*) indicate significant differences compared to WT (Student’s *t* test; * *p* < 0.05; ** *p* < 0.01).

## Data Availability

The datasets generated during and/or analysed during the current study are available from the corresponding author on reasonable request.

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
