# Peer review of "Modified Crosstalk between Phytohormones in Arabidopsis Mutants for PEP-Associated Proteins"

_ijms, 2024, doi:10.3390/ijms25031586_

Round 1
Reviewer 1 Report
Comments and Suggestions for Authors
In this manuscript, the authors conducted phytohormone determination, transcript analysis, and assessed phytohormone responses in two Arabidopsis mutants lacking PEP-associated proteins (PAPs) to examine the effects of plastid development-associated crosstalk between phytohormones. The mutants exhibited both up- and down-regulated gene expression in several phytohormone synthesis and degradation pathways, along with alterations in various phytohormone levels. Additionally, the mutants displayed distinct sensitivity to various phytohormones. Based on these observations, the authors proposed that plants maintain a balance between phytohormone signaling and development to ensure viability under different genetic backgrounds.
While the manuscript provides comprehensive data, it primarily remains descriptive, offering limited assistance in understanding the intricate signaling network involved in plastid metabolisms. The following specific points need clarification by the authors:
1) The authors should include schematic models illustrating the chloroplast-derived signaling network they propose, akin to Figure 3 in reference 6, to enhance readers' comprehension.
2) Although raw data are presented in supplemental data (Tables S1, S2, S3, and S4), crucial information about experimental replicates is lacking. For instance, the authors should specify how many times each experiment was conducted, and whether these were biological and/or experimental replicates. Clarifying these details will enhance the transparency and reliability of the presented findings.
Author Response
We are grateful to the reviewer for the careful analysis of the manuscript and the comments made and would like to respond to some of them.
- The authors should include schematic models illustrating the chloroplast-derived signaling network they propose, akin to Figure 3 in reference 6, to enhance readers' comprehension.
Inactivation of PAP1 and/or PAP6 nuclear genes encoding proteins that together with bacterial-type plastid RNA polymerase participate in the formation of the chloroplast transcription complex leads to albino phenotype. In such plants, a powerful retrograde signal acts, which promotes suppression of nuclear gene expression, mainly involved in the photosynthesis. Chloroplasts contain 350-400 proteins participating in many metabolic processes of the cell, but with the inactivation of chloroplasts, metabolic pathways will also be altered. Plants deprived of chlorophyll experience light stress even under optimal growth conditions. Some steps of biosynthesis of a number of phytohormones (e.g. cytokinins, ABA and others) occur in chloroplasts. Therefore, the balance of phytohormones would be altered in such mutants due to stress and metabolic changes. According to our results, the balance of the major phytohormones and their precursors and metabolites was significantly altered in albino plants. The content of phytohormones suppressing chloroplast biogenesis (ABA and MJ) decreased, while the content of cytokinin (or rather its derivatives) activating biogenesis increased (our previously published article Andreeva et al., Biomolecules 2020, 10, 1658. https://doi.org/10.3390/biom10121658). The level of auxins was almost unchanged, but the content of modified auxins increased. Quite unexpectedly, there was a significant increase in the content of salicylic acid. The change in the balance of phytohormones in the mutants that we detected probably provided better survival and development of the mutants under heterotrophic nutrition. In our opinion, this balance of phytohormones was achieved by altering phytohormone biosynthesis.
The situation is quite different with hormone signalling. So far, not a single receptor or component of the hormonal signal transduction pathway has been found for any of the phytohormones among the 350-400 chloroplast proteins. After sequencing thousands of chloroplast genomes, it became quite clear that they could not encode any components of hormone signalling due to a lack of genetic information. In our opinion, chloroplast gene expression is mediated via nuclear encoded hormone receptors. Thus, chloroplasts stand somewhat apart from hormone signalling.
We did not plan to study the changes in the expression of the components of phytohormone signalling, as this is a vast independent work that is not really justified in connection with chloroplasts. However, we could not help but test whether the perception of hormone signals was impaired in the mutants. To check this, the expression of primary response genes to individual phytohormones is usually studied. We did this and obtained an important answer that phytohormone signalling is not impaired in the chlorophyll-free mutants of A. thaliana.
Based on the above, we cannot propose a scheme for hormone signalling involving chloroplasts. However, we have inserted a schematic diagram (Fig. 5) illustrating the major shifts in the content of various hormones arising from dysfunctional chloroplasts, which can be interpreted as a consequence of retrograde signaling.
- Although raw data are presented in supplemental data (Tables S1, S2, S3, and S4), crucial information about experimental replicates is lacking. For instance, the authors should specify how many times each experiment was conducted, and whether these were biological and/or experimental replicates. Clarifying these details will enhance the transparency and reliability of the presented findings
We have added lacking information to supplementary data.
Reviewer 2 Report
Comments and Suggestions for Authors
The manuscript of Bychkov et al. entitled "Modified Crosstalk between Phytohormones in Arabidopsis Mutants for PEP-Associated Proteins” represents an interesting contribution in the understanding of the retrograde signaling in plants and the intricacies of the signalling. The paper is generally well written though a thorough reviewing was not possible because the text is not numbered. How to properly review a MS in these conditions?
However, there are a few things that have to be reviewed, most of them around the number of biological replicates used in the experiments and the way statistics was conducted and the age of the plants.
Page 2
Modify the size of the bracket and remove the marking of 1-aminocyclopropane-1-carboxylic acid.
“(GA3), brassinolide (BL) or the ethylene precursor 1-aminocyclopropane-1-carboxylic acid ….”
Figure 1, pages 3-4.
“(A) Phenotypes of pap mutants and wild-type plants grown on Murashige and Skoog (MS) media in Petri dishes for 4 weeks.”
These are not photos of 4 weeks old plants. It is impossible. These are 10 days, maybe two weeks old plants. It would be interesting to see some 4 weeks old plants on MS media, especially of the mutants!
Indicate the number of biological replicates. The only information provided in section 4.1: “Leaves from at least six different plants were harvested for each probe to average the results.” What is a probe? A biological replicate? How many biological replicates have been used? Three? If yes, there were 3 replicates, and each replicate contained plant material from 6 plants.
Figure 2. Page 6
How was the normalization of the qPCR data done. WT was considered as 1? How were the grouping letter assigned. Usually, the highest value will be a. Why the highest value is, in most cases, b? But for some genes, for example ICS1, c is the highest value. Statistical processing was not done properly. Please review the way letter grouping was done and do it in a consistent manner!
Figure 4. Page 10.
The ratio in the “MS” column should be written with a dot not a comma, that is, 1.5, not 1,5.
Page 13
4.2
Different size of the fonts
Applied Biosystems, Waltham, MA, USA).
4.3
Why “Roche, Roche”?
4.4
Protein concentrations are not expressed in μg/mm2! Maybe μg/μl or μg/ml or μg/mm3
Comments on the Quality of English Language
Good quality.
Author Response
Thank you so much for your review and helpful comments. We apologize for the unnumbered text, which made it difficult to review. We have made corrections to the manuscript and tried to respond to some comments.
Page 2
Modify the size of the bracket and remove the marking of 1-aminocyclopropane-1-carboxylic acid.“(GA3), brassinolide (BL) or the ethylene precursor 1-aminocyclopropane-1-carboxylic acid ….”
Fixed.
These are not photos of 4 weeks old plants. It is impossible. These are 10 days, maybe two weeks old plants. It would be interesting to see some 4 weeks old plants on MS media, especially of the mutants!
I would like to note that the ontogeny of pap mutants is extremely extended and significantly lags behind wild-type plants. So, in Fig. 2B only the cotyledons are visible in 11-day-old seedlings (11 days after germination or 14 days after soaking the seeds), and the first true leaves just begin to appear. As for wild-type plants, after three weeks on MS medium they also retard in their growth and look significantly younger than soil plants, which by this time have already begun bolting. As a supplement, we have attached photograph of 7-week-old pap1 mutant (Fig. 5) illustrating the consequences of blocking chloroplast biogenesis.
Indicate the number of biological replicates. The only information provided in section 4.1: “Leaves from at least six different plants were harvested for each probe to average the results.” What is a probe? A biological replicate? How many biological replicates have been used? Three? If yes, there were 3 replicates, and each replicate contained plant material from 6 plants.
Fixed
Figure 2. Page 6
How was the normalization of the qPCR data done. WT was considered as 1? How were the grouping letter assigned. Usually, the highest value will be a. Why the highest value is, in most cases, b? But for some genes, for example ICS1, c is the highest value. Statistical processing was not done properly. Please review the way letter grouping was done and do it in a consistent manner!
Yes, for ease of understanding of the figures, values for wild-type control plants are set to 1. The grouping of letters has been corrected.
Figure 4. Page 10.
The ratio in the «MS» column should be written with a dot not a comma, that is, 1.5, not 1,5.
Fixed
Page 13
4.2
Different size of the fonts
Applied Biosystems, Waltham, MA, USA).
Fixed
4.3
Why “Roche, Roche”?
Fixed
4.4
Protein concentrations are not expressed in μg/mm2! Maybe μg/μl or μg/ml or μg/mm3
Indeed, protein content is usually expressed as μg/μl of probe or in μg per mg of fw. However, when using the senescence test with dark-detached leaves, more accurate results are obtained by calculating the protein or chlorophyll content per unit of leaf-blade area, since various samples may lose weight differently in course of incubation, while the leaf area remains stable (Burkhanova et al, Plant growth Regulation, 33, 195-198,2001).
Round 2
Reviewer 2 Report
Comments and Suggestions for Authors
The authors addressed by comments.